# Surface Plasmon Resonance Biosensor Chip for Human Blood Groups Identification Assisted with Silver-Chromium-Hafnium Oxide

**Purnendu Shekhar Pandey** [1] , **Sanjeev Kumar Raghuwanshi** [2,*], **Rajesh Singh** [3,4] and **Santosh Kumar** [5,*]

1 Department of Electronics Engineering, GL Bajaj Institute of Technology and Management, Greater Noida 201306, India
2 Department of Electronics Engineering, Indian Institute of Technology (Indian School of Mines) Dhanbad, Dhanbad 826004, India
3 Uttaranchal Institute of Technology, Uttaranchal University, Dehradun 248007, India
4 Department of Project Management, Universidad Internacional Iberoamericana, Campeche C.P. 24560, Mexico
5 Shandong Key Laboratory of Optical Communication Science and Technology, School of Physics Science and Information Technology, Liaocheng University, Liaocheng 252059, China
* Correspondence: sanjeevrus77@iitism.ac.in (S.K.R.); santosh@lcu.edu.cn (S.K.)

**Abstract:** Chromium (Cr), silver (Ag) and hafnium oxide ($HfO_2$) are used in a surface plasmon resonance (SPR)-based biosensor with an optimized design for measuring blood groups at a wavelength of 633 nm. A buffer layer was placed on the SPR active metal in this investigation to avoid oxidation and contamination of blood samples. A theoretical model based on experimental data considered the refractive index of blood samples. The BK7 prism is the optimum substrate material for blood type identification analysis using a combination of Ag and Cr as an SPR active metal. The sensor's performance is carefully researched in terms of its angular shift and curve width to predict the design aspects that provide precise blood-group identification. The SPR dip slope, detection accuracy and figure of merit (FOM) have been investigated concerning the subsequent generation of biosensor applications.

**Keywords:** surface plasmon resonance; sensitivity; bimetallic; blood sample; biosensors

## 1. Introduction

The Surface plasmon resonance sensor has received much attention among sensing techniques because of its high sensitivity and wide range of applications. Surface plasmon resonance (SPR) effects are not just for sensory purposes. Applications for optoelectronic devices such as optical tunable filters were discovered by researchers [1,2], along with applications as modulators [3,4], thin-film thickness monitors [5], SPR images [6] and liquid sensors [7,8]. Ritchie was the first to conceptually introduce the phenomenon of surface plasmons (SPs) [9].

SPR sensors require a metallic component that has a significant number of free electrons. These free electrons provide the real part of the negative permittivity required for plasmonic materials. The SPP wave propagates along the prism surface in the typical Kretschmann setup when p-polarization or transverse magnetic light strikes the prism and is coated with a plasmonic material (Ag, Au, Cu) [10]. The propagation constant of the SP mode changes when the dielectric refractive index (RI) changes. As a result, the coupling state between the SP and light wave varies and may be monitored using the optical wave connecting with the SP mode characteristics [11,12]. When treating patients who have suffered massive blood loss, it is necessary to identify patients' blood groups. In order to prevent an instance of blood incompatibility during the transfusion process, it is necessary to ensure that the blood group of the patient and the donor are compatible. Because of its powerful specific antigen-antibody interactions, the "A", "O" and "B" blood typing system

is always checked first for all blood transfusions. This is because the "A", "O" and "B" blood identifying system has the potential to cause severe damage to all other systems [13].

Performance has been studied theoretically and the fabrication prospects with the proposed structure discussed. The deposition of the materials has been carried out on silica glass and pasted with the refractive index matching gel, and reflectance has been investigated. A brief summary describes the selection of materials in Section 2 and mathematical discussion and fabrication prospects are given in Section 3.

## 2. SPR Sensor Theoretical Modeling and Design Considerations

Figure 1 shows the four-layer proposed structure for the blood group measurement of "A", "O" and "B". The proposed novel heterostructure using the dielectric buffer layer (DBL) has been investigated for blood groups' "A", "O" and "B" measurement. The brief introduction of each layer used in the proposed biosensor is:

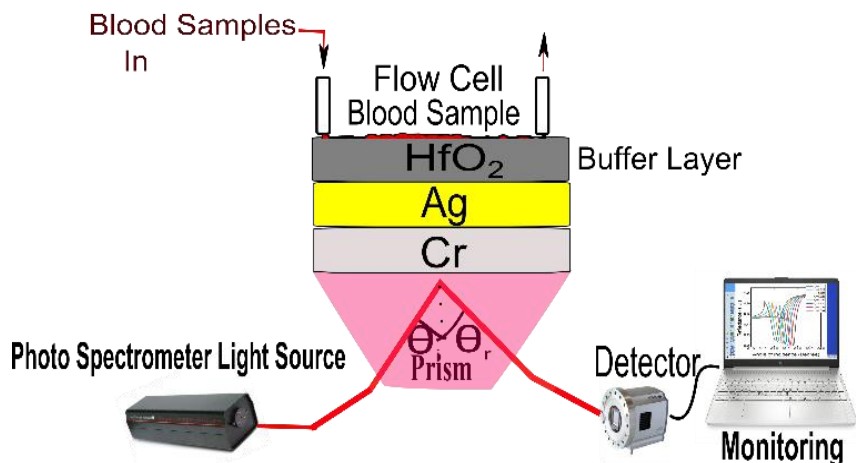

**Figure 1.** Schematic of the proposed prism-based S.P.R. biosensor for identifying human blood groups.

1.   **BK7 Prism selection**

BK7 is a borosilicate crown glass that complies with the standards. It has high transmission in visible and near-infrared regions (350–2000 nm). BK7 is almost certainly the type of optical glass utilized most frequently in the production of high-quality optical components. The tough glass known as BK7 is resistant to a wide range of corrosive conditions, both chemical and physical. It is resistant to chemicals and scratches to a certain extent. As a result of its low bubble and inclusion content, it is an excellent choice for fabricating precision lenses. It has a RI of 1.5151, indicating that it is highly reflective. The BK7 glass refractive index can be calculated using the following Equation (1) [14]:

$$n_{BK7} = \left( \frac{a1\lambda^2}{\lambda^2 - b1} + \frac{a2\lambda^2}{\lambda^2 - b2} + \frac{a3\lambda^2}{\lambda^2 - b3} \right) \tag{1}$$

Table 1 shows the constant parameters required to calculate the refractive index of the BK7 prism as per Equation (1).

**Table 1.** As given in the Equation (1), the value of the constant's parameters.

| $a_1$ | $a_2$ | $a_3$ |
|---|---|---|
| 1.03961212 | 0.231792344 | 1.03961212 |
| $b_1$ | $b_2$ | $b_3$ |
| 0.00600069867 $\mu m^2$ | 0.0200179144 $\mu m^2$ | 103.560653 $\mu m^2$ |

## 2. Silver (Ag) metal selection

In order to excite surface plasmons that can be utilized in biosensors, a nanometer-thick layer of a metallic substance, such as silver or gold, is deposited on the prism in the form of a thin layer [15]. Due to their high signal-to-noise ratio (SNR) and strong SPR signals, silver (Ag) and gold (Au) have been extensively explored as metal films for nonmaterial. As a result, silver (Ag) is the metal that is best suited for SPR applications. Silver, on the other hand, oxidizes much more rapidly than other metals due to its low chemical and optical stability. Because of its low capacity for binding to biomolecules and broad SPR curve, gold-based biosensors have lower sensitivity than silver-based equivalents [16]. Because of its superior oxidation resistance and chemical properties, gold is well suited for use in sensing applications [17]. Compared to Au films, Ag films typically have a more prominent peak and improved sensitivity, making them an excellent choice for increasing the sensitivity of biosensors [15]. For the proposed structure, the thickness of Ag layer was chosen as 50 nm for achieving maximum sensitivity, as shown in Figure 2. Table 2 illustrates the Au and Ag sensitivity obtained and demonstrates why Ag metals are chosen in the proposed structure.

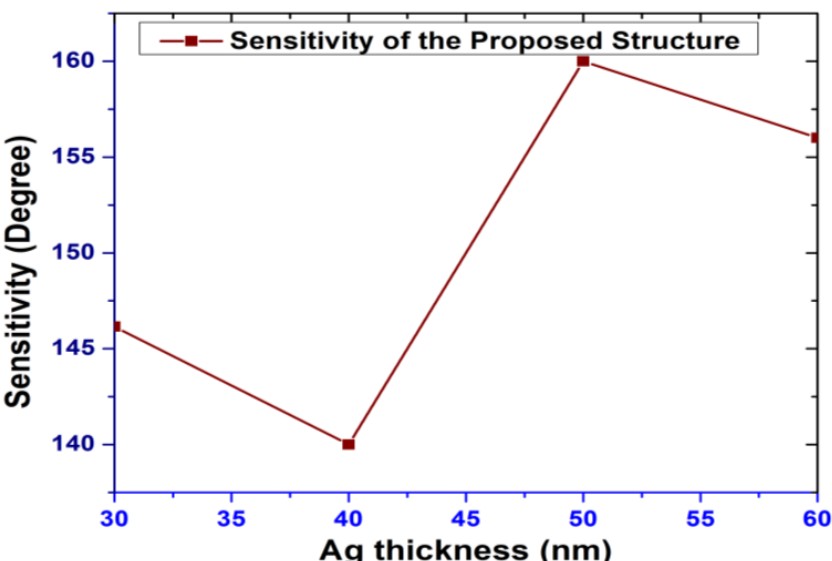

**Figure 2.** Sensitivity for the Ag layer thickness from 30 nm to 60 nm.

**Table 2.** Au and Ag thickness comparison for sensitivity of the proposed structure.

| Au Thickness (nm) | Sensitivity (°/RIU) | Ag Thickness (nm) | Sensitivity (°/RIU) |
|---|---|---|---|
| 30 | 136.41 | 30 | 146.15 |
| 40 | 148.91 | 40 | 140 |
| 50 | 158.21 | 50 | 160 |
| 60 | 151.23 | 60 | 156 |

## 3. Chromium (Cr) layers

In addition, for Kretschmann-based SPR sensing, an additional adhesive layer is required to keep the metallic layer in solid contact with the prism. Additional atoms such as oxygen and hydrogen typically cause poor adhesion between the metallic layer and the prism. Some articles provides direct evidence that the primary cause of roughness in polycrystalline metal films is the island size just before coalescence [18]. Using the polarimetric technique, an investigation into the optical properties of chromium is described. Researchers have looked at the optical properties of very thin chromium films and whether

or not they can be used in scanning reflection interferometry. The range of thickness was from 0 to 36 nm. An absolute reflectometer was used to measure the reflectance and transmittances [19]. As an oxygen-active element, chromium forms highly stable nucleation centers on glass or silicon oxide. It also possesses a high degree of toughness and corrosion resistance. While the influence of the adhesion layer can be disregarded in most applications, it has a substantial effect on the near-field response of plasmonic sensors to plasmon resonance.

### 4. Hafnium oxide ($HfO_2$)

Hafnium oxide possesses chemical stability, a high dielectric constant (20–25), a wide band gap (5.8 eV), a conduction band offset (1.4 eV), optical transparency from 300 to 10,000 nm in the electromagnetic spectrum and a RI (~2) that is dependent on deposition conditions ($HfO_2$) [20]. It is also transparent from around 250 to 900 nm, with a comparatively high refractive index n = 1.91 at $\lambda$ = 632.8 nm. $HfO_2$ is an appealing dielectric for SPPs-based sensors due to substantially greater values of the dielectric constant, melting temperature, density and range of wavelengths over which the material is optically transparent. When used as a gate dielectric for charge-based biosensors, $HfO_2$ demonstrated durability in an aqueous electrolyte environment and biotin biomolecule probes were successfully functionalized on it. Temperature, pressure, voltage, plasma composition and annealing are deposition parameters that influence $HfO_2$ film properties. The $HfO_2$ layer is deposited using electron beam evaporation as a buffer layer. Due to its chemical and biological properties, a single biochemical layer can provide stable bonding at the two interfaces (Ag layer and blood sample) to prevent structural anomalies (e.g., POLYETHYLENE GLYCOL or thiol). For sensitive SPR measurements, the biological layer should be 1–15 nm thick on silica substrates, according to studies. Here, in the proposed structure, we have coated 30 nm of $HfO_2$ and numerically analyzed it at 30 nm SPR angle maximum.

### 3. Results and Discussion

The proposed structure of the SPR sensor is considered with the Kretschmann configuration. The proposed sensor is a five-layer (BK7/Cr/Ag/$HfO_2$/blood sample) structure shown in Figure 1. At one face of the prism, transverse magnetic (TM) polarized light from the source with an operating wavelength ($\lambda$) of 633 nm is applied, and the reflected light is obtained using appropriate photodetector array instruments. A thin nano order of the thickness of metallic layers such as Silver (Ag) is deposited as a first layer on the prism for excitation of SPs for use in surface plasmon biosensors [16]. The thickness and RI of the Ag layer was chosen as 50 nm and $0.0562 + 1i \times 4.2776$, respectively. For further excellent oxidation resistance and chemical stability, an Ag layer is also deposited over the Cr. Furthermore, for Kretschmann-based SPR sensing, an extra adhesive layer is necessary to maintain solid contact between the metallic layer and the prism [21]. Ag was chosen for the second layer, above the Cr. The RI of the Cr layer is $3.1395 + 1i \times 3.3152$ and the thickness taken is 10 nm.

Additional atoms such as oxygen and hydrogen between the metallic layer and the prism usually cause poor adhesion. Some articles provide direct evidence that island size shortly prior to coalescence is the primary cause of roughness in polycrystalline metal films [22]. Because chromium is an oxygen-active substance, it forms stable nucleation centers on glass or oxidized silicon. The third layer is Hafnium oxide ($HfO_2$) as a buffer layer (BL). The refractive index of $HfO_2$ is taken as 2.042 with a thickness of 30 nm [23]. At 633 nm, for blood group A, a refractive index ($n_A$ = 1.3739), for blood group B ($n_B$ = 1.3783) and for blood group O ($n_O$ = 1.3778) is considered [24]. Figure 3 shows the reflectance graph with respect to angle of incidence RI values of blood groups A, B and O for the proposed geometry. Table 3 shows the parameters selected to analyze the proposed structure's SPR performance, as shown in Figure 1. The RI of the blood samples is obtained in order of $n_A < n_O < n_B$. The maximum sensitivity for the proposed structure was obtained as 160°/RIU. D.A. for blood samples "A", "O" and "B" obtained 0.75188, 0.724638 and

0.714286, respectively. Furthermore, the FOM for analyzed blood samples "A", "O" and "B" showed RI of 1.33, 1.38 and 1.4, respectively.

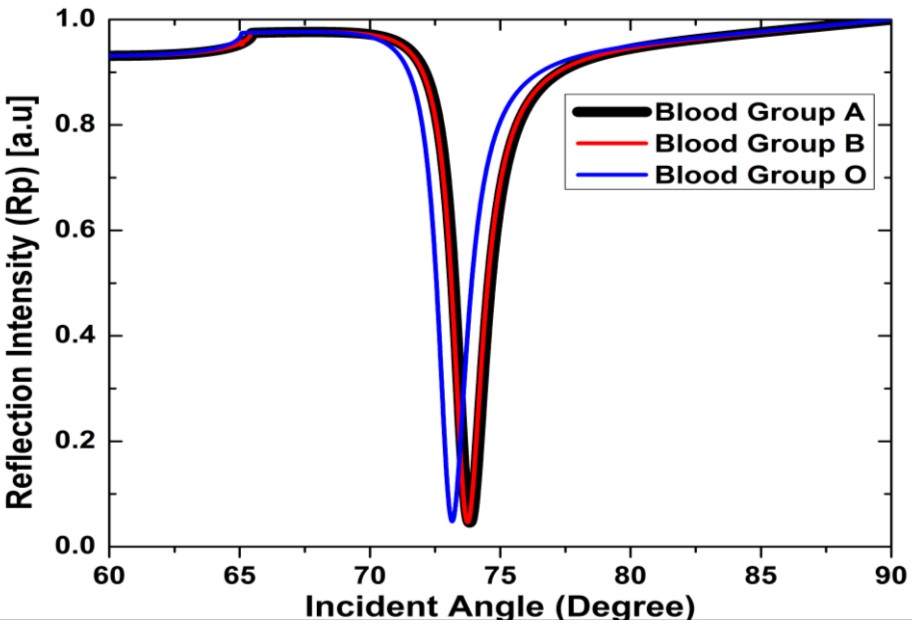

**Figure 3.** Angle of Incident at a refractive index value of blood groups "A", "O" and "B".

**Table 3.** Design parameters of the proposed sensor at 633 nm.

| Layers | Materials | Refractive Index | Thickness (nm) |
|---|---|---|---|
| I | BK7 Prism | 1.5151 | - |
| II | Ag | $0.0562 + 1i \times 4.2776$ | 50 nm |
| III | Chromium | $3.1395 + 1i \times 3.3152$ | 10 nm |
| IV | $HfO_2$ | 2.042 | 30 nm |
| V | Sensing layer | $n_A = 1.3739$, $n_B = 1.3783$, $n_O = 1.3778$ | - |

Coupling equation for incident light and SPs at the metal–dielectric interface, as shown in Equation (2);

$$K_z = K_{sp} = \frac{2\pi}{\lambda} n_p \sin\theta_{SPR} K_z = K_{sp} = \frac{2\pi}{\lambda} n_p \sin\theta_{SPR} \text{real}\left(\frac{2\pi}{\lambda}\sqrt{\frac{\varepsilon_m \varepsilon_s}{\varepsilon_m + \varepsilon_s}}\right) \qquad (2)$$

In Equation (2), $n_p$ refers to a RI of the substrate medium and lambda ($\lambda$) is the light wavelength. $\varepsilon_m$ and $\varepsilon_s$ are the dielectric constants of the metal layer and the sensing (analyte) layers, respectively. The coupling of incoming light and SPs at the metal–dielectric contact is likewise represented by the Equation (1). Sensitivity is defined by Equation (3):

$$S = \frac{\Delta\theta_{spr}}{\Delta n_s} \qquad (3)$$

and detection accuracy (DA) is defined by Equation (4):

$$DA = \frac{1}{FWHM} \qquad (4)$$

The field associated with surface plasmon wave (SPW) is expressed as:

$$E = E_o \, e^{\,i(k_x X \pm k_z Z - \omega t)} \tag{5}$$

In Equation (5) signs of + and − are represented for $z \geq 0$ and for $z \leq 0$, respectively. Equation (5) represents the exponential decay of the field. Wave vector ($k_x$) is parallel to the *x*-axis and is given as:

$$k_x = \frac{2\pi}{\lambda p} \tag{6}$$

where $\lambda_p$ is plasma wavelength. Dispersion relation of SPW can be obtained by Equation (5) by applying the boundary condition.

Figure 4 shows Sensitivity, FOM and Incidence angle with respect to RI for blood samples "A", "O" and "B". $\theta_{SPR}$ at blood groups "A", "B", and "O" was 73.15°, 73.75°, and 73.83°, respectively.

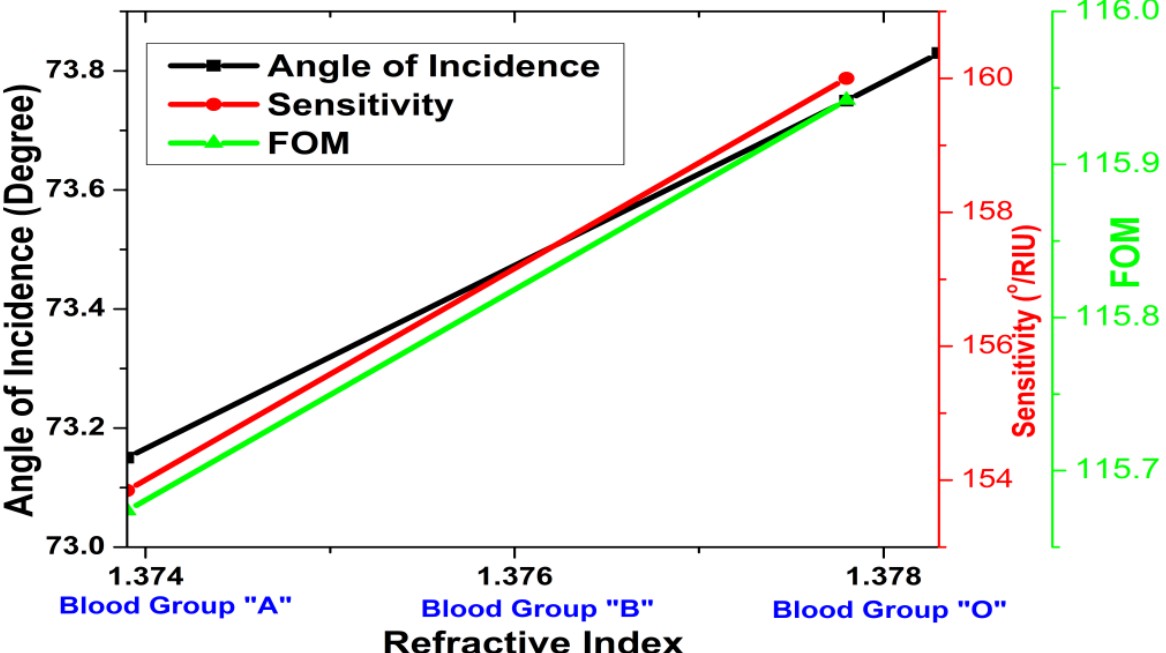

**Figure 4.** Proposed S.P.R. performance parameters (F.O.M, Sensitivity, Incident Angle) with respect to RI of Blood group samples "A", "O" and "B".

Figure 5 shows minimum reflectance, FWHM and DA of the proposed SPR sensor for RI of blood sample "A", "O" and "B". It is reported that at RI of blood group B the FWHM is high and achieved minimum reflectance.

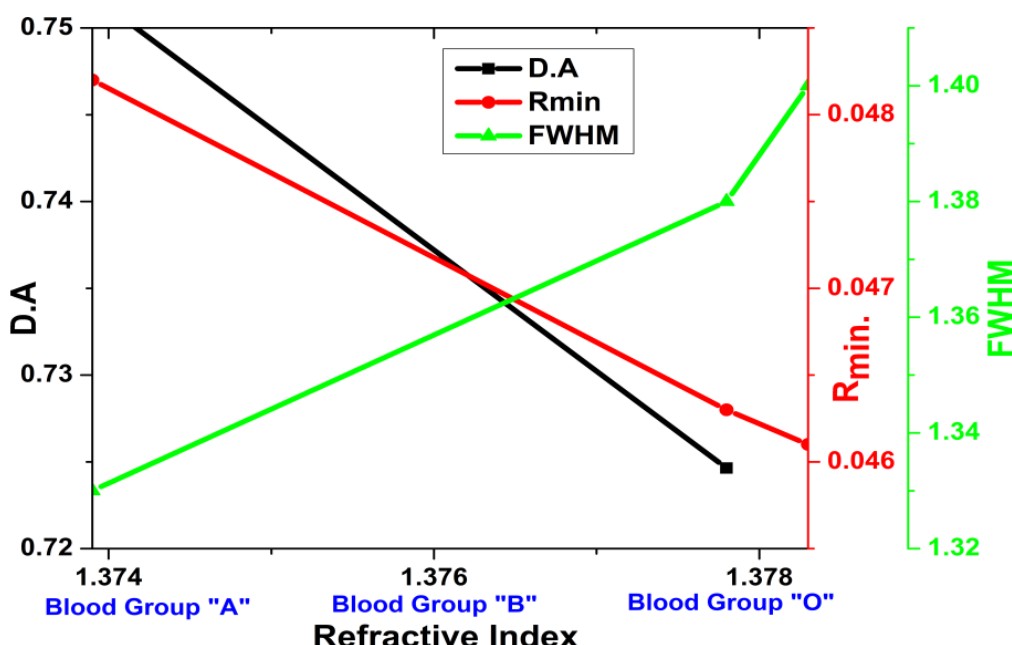

**Figure 5.** Minimum Reflectance, FWHM and DA of proposed SPR sensor with respect to RI of Blood group samples "A", "O" and "B".

### 4. Fabrication Prospects

After the theoretical analysis, this section discusses experimental fabrication prospects for blood group measurement. We fabricated the structure with the available prism (THOR LAB, PS991) and Silver (Ag), Chromium (Cr) and $HfO_2$ (Hafnium oxide). The proposed structure (BK7/Cr/Ag/$HfO_2$/Blood sample)-based SPR biosensor is as shown in Figure 1. Low bubble and inclusion content makes it ideal for making precision lenses. Figure 6 shows the layer-by-layer deposition of $HfO_2$, Cr and Ag on the silica glass.

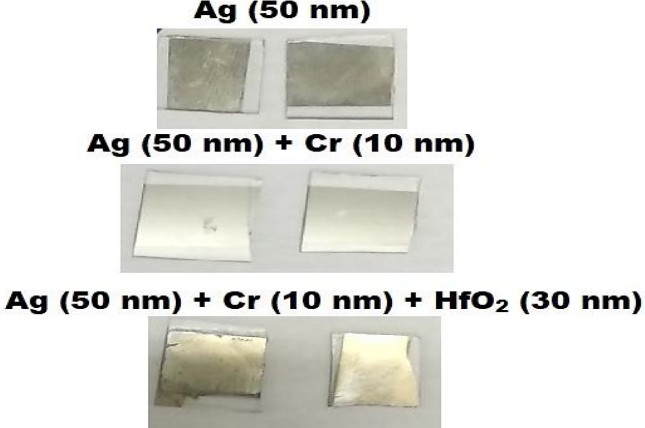

**Figure 6.** Stepwise coating for Ag, Cr and $HfO_2$ on Silica glass.

Figure 7 shows the experimental setup of the blood group measurement. Taking 5 mL of blood group samples A, B, O and with the help of an optical source, the reflectance (Spetrophotometer, New Age Instrument and materials Pvt. Ltd. ERA Si) was investigated sample by sample. Deposition on silica glass of Ag (10 nm), Cr (10 nm) and $HfO_2$ (30 nm) has been performed with the help of an electron beam gun/system as shown in Figure 8.

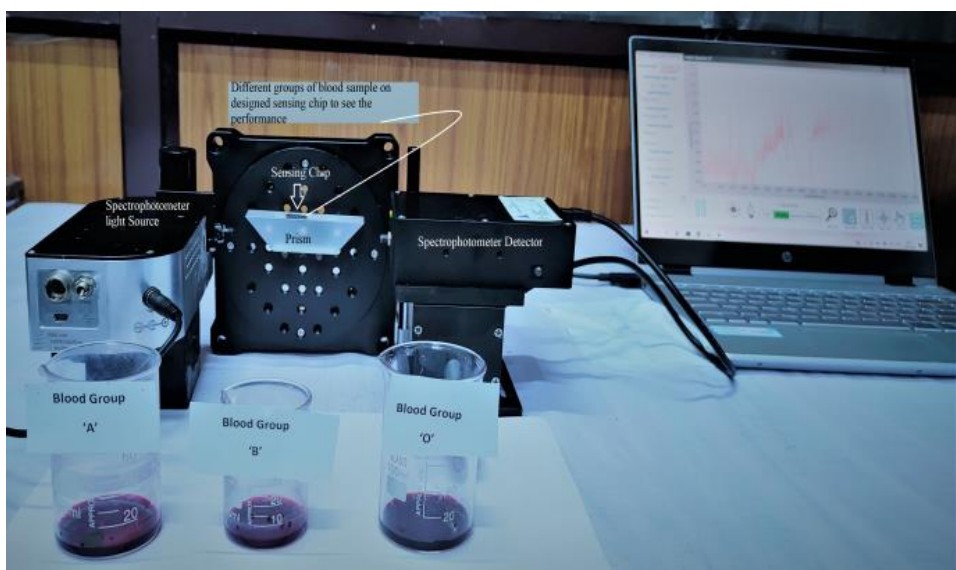

**Figure 7.** Prism based experimental set-up for blood group identification integrated with photo spectrometer.

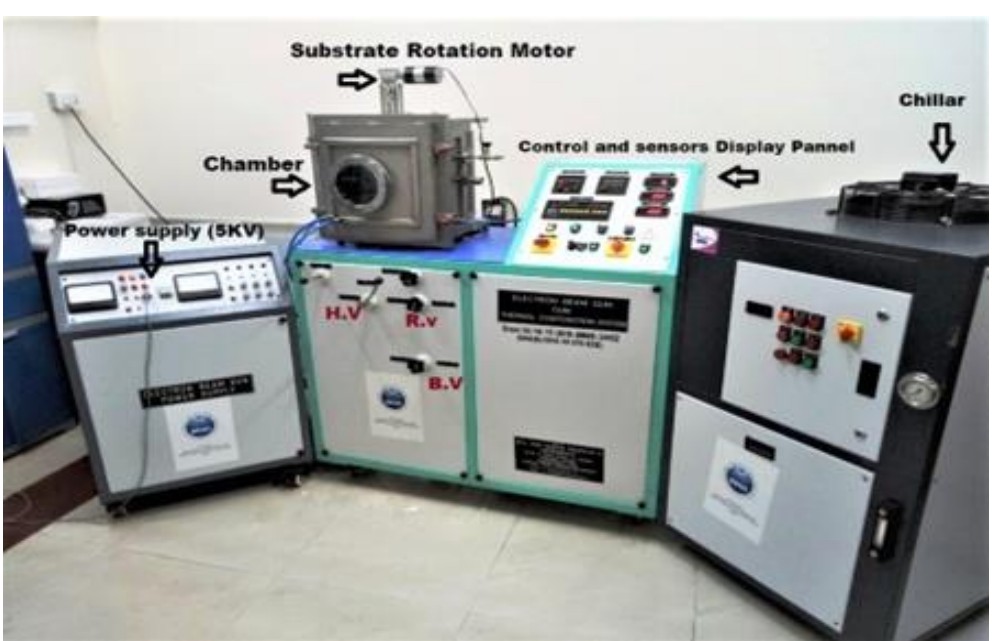

**Figure 8.** Complete set up of Deposition Machine.

Ag, Cr and $HfO_2$ materials were used to coat silica glass by a process using the electron beam gun deposition system as discussed above. The sensing chip is developed by a multilayered structure coated onto the high refractive glass with the desired materials. The BL layer is deposited over the glass substrate using e-beam evaporation. The produced SPR chip is then applied to the flat surface of the prism with index matching gel. The sensing analyte was transferred to the flow cell on the sensing chip's upper surface. During the coating process, the electron beam gun is maintained at 5 kV high voltage with vacuum chamber pressure of $10^{-6}$ mBarr. The beam from the electron gun directly hits the material. The material starts to melt, then evaporating and starting to coat the prism, the coating rate controlled by varying the current knob, and monitoring of the thickness is carried out by a digital thickness monitor [DTM], at a preferred coating rate of 0.3 KA/s. After obtaining the required thickness over the prism, the shutter is closed and the thickness is

noted and the coating material are noted for verification, using field emission scanning electron microscopy (FESEM) as shown in Figure 9.

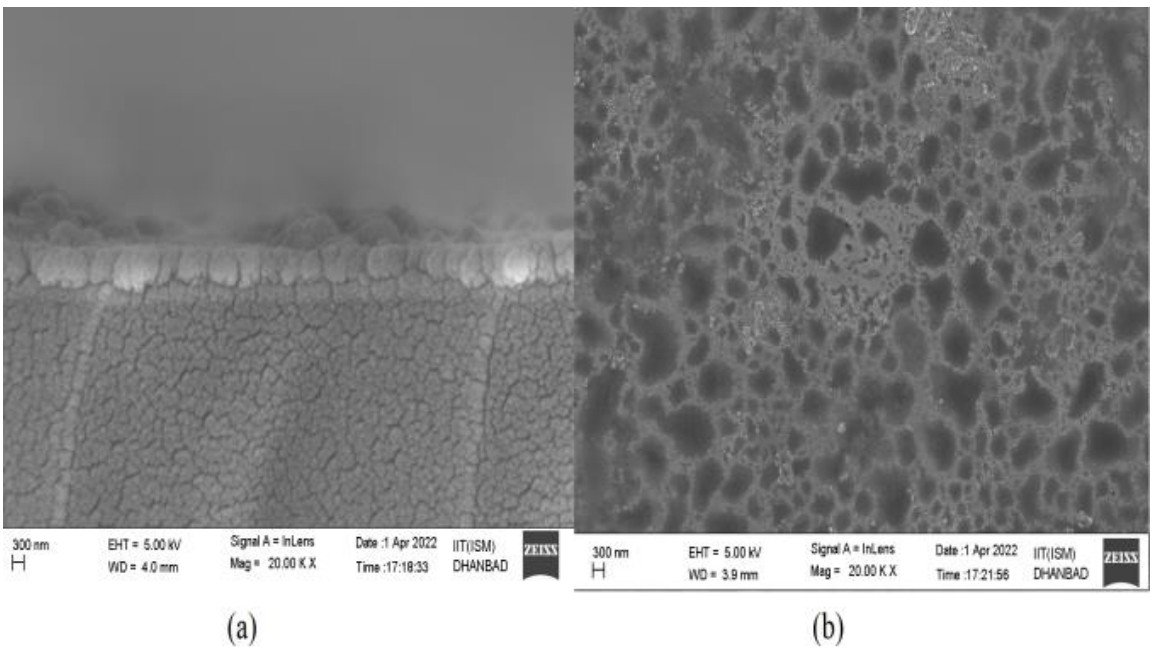

**Figure 9.** (**a**) Cross-sectional view and (**b**) top view of the deposited material on sample through FESEM image.

The pressure range maintained by the system during the experiment is defined as low at 760 Torr and as outer space at $\approx 10^{-16}$ Torr, but the available system can only work until $10^{-6}$ millibars, which is sufficient for the performance of any laboratory experiment and for coating the required material over the substrate as in Table 4.

**Table 4.** VACUUM quality with respect to pressure ranges.

| Types of Vacuum | Pressure Range (Torr) |
| --- | --- |
| Low | 760–0 |
| Medium | $0$–$10^{-3}$ |
| High | $10^{-3}$–$10^{-8}$ |
| Outer Space | $\approx 10^{-16}$ |

After deposition of materials, FESEM is used for the measurement of the thickness as shown in Figure 9. Figure 9a shows the cross-sectional view of materials (Ag, Cr and HfO$_2$) deposited on silica glass to see the film thickness of the coating materials and Figure 9b shows the top view the coated materials.

**5. Conclusions**

In the current investigation, a method for identifying blood groups is presented. This method is based on the Kretschmann configuration of an SPR sensor. Oxidation and other difficulties related to direct physical contact of blood samples with SPR active metal are no longer a concern in the ongoing experiment now that a buffer layer (HfO$_2$) has been added on top of the SPR active thin metal film. The performance of the sensor is extensively evaluated in terms of its angular shift and curve width in order to provide accurate predictions regarding the design aspects that will be required to enable accurate blood-group identification. The transfer matrix technique was applied to the analysis of

actual experimental data obtained from blood samples "O", "A" and "B". A study of the performance of an SPR biosensor for blood type detection was conducted in terms of blood discrimination factor ($\delta_{\theta SPR}$), FOM, DA and sensitivity. Theoretically, SPR performance has been investigated and fabrication prospects are given in detail. The proposed structure can be useful for rapid detection of blood group samples "O", "A" and "B".

**Author Contributions:** P.S.P. work on conceptualization and analysis for this research work. S.K.R. contributed for the investigation and experimental data validation of blood samples for the proposed sensor. R.S. contributed to add methodology. S.K. contributed as investigation, validation, writing and editing the manuscript. All authors have read and agreed to the published version of the manuscript.

**Funding:** This work is carried out by Research Grant under project reference no. SCP/2022/000271 and project no. DST (SERB)(356)/2022-2023/955/ECE dated 8 August 2022 funded by Science and Engineering Research Board, Department of Science and Technology, Government of India with the project entitled *Design of a web server-based hybrid physiological sensor with optical cloth for real-time health specialist care.* The work of Santosh Kumar was supported by the Double-Hundred Talent Plan of Shandong Province, China.

**Institutional Review Board Statement:** Not applicable.

**Informed Consent Statement:** Not applicable.

**Data Availability Statement:** Not applicable.

**Acknowledgments:** This work is carried out by Research Grant under project reference no. SCP/2022/000271 and project no. DST (SERB)(356)/2022-2023/955/ECE dated 8 August 2022 funded by Science and Engineering Research Board, Department of Science and Technology, Government of India with the project entitled *Design of a web server-based hybrid physiological sensor with optical cloth for real-time health specialist care.* The work of Santosh Kumar was supported by the Double-Hundred Talent Plan of Shandong Province, China.

**Conflicts of Interest:** The authors declare no conflict of interest.

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
