# Peer review of "Surface Plasmon Resonance Biosensor Chip for Human Blood Groups Identification Assisted with Silver-Chromium-Hafnium Oxide"

_magnetochemistry, doi:10.3390/magnetochemistry9010021_

Round 1
Reviewer 1 Report
This manuscript describes the estimation of blood type using a surface plasmon sensor. In order to accept it, I think that the following points should be corrected.
It should show the novelty of this sensor. Much research has been done on surface plasmon sensors in the Kretschmann configuration, and it should be elaborated where the novelty lies for blood group estimation.
I don't think it's clear why you chose the materials that make up the sensor.
・Why is BK7 used as a prism material? For example, why can't it be used with quartz glass?
・Cr is used to improve the adhesion between Ag and BK7, but the effect of Cr on optical properties is not clearly stated.
・Ag has a problem of corrosion, but what is the reason for using Ag?
・In Figure 2, the Ag film thickness seems to change the sensitivity. Is this a significant change? It is possible that it cannot be seen as a noticeable change. From this result, can we conclude that 50nm is the appropriate film thickness?
・HfO2 is used as a buffer layer, but is it correct to assume that this acts as a protective film against corrosion of Ag? What makes HfO2 better than other materials?
In Figure 3 the reflectance spectrum of Blood group of A is not shown.
In Figures 4 and 5, it seems unreasonable to connect the polygonal lines with a small number of measurement points. You should draw a graph with more measurement points.
Author Response
Thanks for your comments. Response file is attached in a pdf file.

Reviewer 2 Report
The manuscript described a surface plasmon resonance-based biosensor for blood type identification. The biosensor contained multiple layers, including BK& Prism, chromium, silver, and hafnium oxide. Blood samples were loaded onto a flow cell on top of the hafnium oxide layer. A polarized light with a wavelength of 633 nm was used as a light source, and the reflected light was obtained using a photodetector. The manuscript contains theoretical discussion, but it lacks strong experimental validation. Therefore, a major revision is suggested.
Comments:
1. It recommended to include blood sample source in the manuscript. I wonder if blood sample pretreatment process is necessary for this work.
2. How many times was the experiment repeated for the data presented in the manuscript?
3. In the second section “SPR sensor theoretical modeling and design considerations”, some arguments are stated without citations. Please include that.
4. In Figure 2, the authors stated, “Compared to Au films, Ag films typically have a more prominent peak and improved sensitivity”. Can the authors elaborate on their data collection? What do they mean by sensitivity in this figure?
5. In line 134, the authors said, “The proposed sensor is a six-layers (BK7/Cr/Ag/HfO2/blood sample) structure 134 shown in Fig. 1”. But there are fiver layers in Figure 1. Please correct it.
6. The “Blood Group A” is not visible on Figure 3.
7. The manuscript stated that Figure 4 and Figure 5 contains blood identification data, but no blood groups are depicted. Can the authors provide additional explanation for the data?
8. In Figure 6, left and right figure in group “Ag (50 nm) + cr (10 nm) + HfO2 (30 nm) are not identical. Can the authors provide an explanation?
Author Response

(The authors gave the same response as above.)

Round 2
Reviewer 1 Report
The authors have made corrections where necessary according to the reviewer's comments, and the manuscript is accepted for publication.
Reviewer 2 Report
The author responded to all comments and made changes to their manuscripts. Acceptance is suggested.